# Conditioning non-linear and infinite-dimensional diffusion processes

**Elizabeth Louise Baker**
Department of Computer Science,
University of Copenhagen
`elba@di.ku.dk`

**Gefan Yang**
Department of Computer Science,
University of Copenhagen
`gy@di.ku.dk`

**Michael L. Severinsen**
Globe Institute,
University of Copenhagen
`michael.baand@sund.ku.dk`

**Christy Anna Hipsley**
Department of Biology,
University of Copenhagen
`christy.hipsley@bio.ku.dk`

**Stefan Sommer**
Department of Computer Science,
University of Copenhagen
`sommer@di.ku.dk`

## Abstract

Generative diffusion models and many stochastic models in science and engineering naturally live in infinite dimensions before discretisation. To incorporate observed data for statistical and learning tasks, one needs to condition on observations. While recent work has treated conditioning linear processes in infinite dimensions, conditioning non-linear processes in infinite dimensions has not been explored. This paper conditions function-valued stochastic processes *without prior discretisation*. To do so, we use an infinite-dimensional version of Girsanov's theorem to condition a function-valued stochastic process, leading to a stochastic differential equation (SDE) for the conditioned process involving the score. We apply this technique to do time series analysis for shapes of organisms in evolutionary biology, where we discretise via the Fourier basis and then learn the coefficients of the score function with score matching methods.

## 1 Introduction

When modelling finite-dimensional data, such as temperature or speed, there are well-known methods for incorporating observations into stochastic or probabilistic models, for example, those based on Gaussian-process regression [Rasmussen and Williams, 2005]. For non-linear models, techniques like Doob's $h$-transform can be used [Rogers and Williams, 2000, Chapter 6]. But for data that is function-valued (and thus infinite-dimensional) with non-linear models, conditioning is still an open problem. This paper introduces a way of conditioning infinite-dimensional diffusion processes by introducing an infinite-dimensional version of Doob's $h$-transform. We then discretise the conditioned process and sample from it; that is, we condition and then discretise rather than discretising and then conditioning.

We present methods to condition a process to hit a specific set at the end time, also known as bridges. This method covers the case of conditioning strong solutions to Hilbert space-valued stochastic differential equations (SDEs). For the conditioning, we consider two scenarios. The first is that the

38th Conference on Neural Information Processing Systems (NeurIPS 2024).

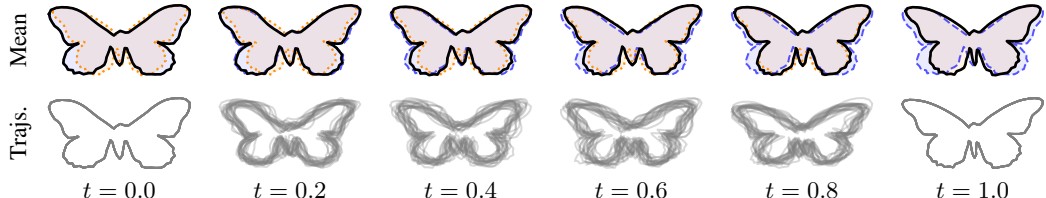

Mean

Trajs.

$t = 0.0$     $t = 0.2$     $t = 0.4$     $t = 0.6$     $t = 0.8$     $t = 1.0$

Figure 1: We condition an SDE between two curves, representing two butterfly species, (starting from red dashed and ending at green dashed). Each time point of the trajectory represents a shape. *Row 1:* We take the mean over 20 trajectories. *Row 2:* We plot the 20 individual trajectories, used in the mean calculation.

transition operators of the SDE solution are smooth, which is generally not obvious [Goldys and Maslowski, 2008]. In the second scenario, we condition on observations with Gaussian noise. This technique can be applied whenever the solution to the SDE is sufficiently differentiable. To condition, we use the infinite-dimensional counterparts of Itô's lemma and Girsanov's theorem, enabling us to define a Doob's $h$-transform analogously to finite dimensions. We then use score-matching techniques, allowing us to sample from the conditioned process. We do this by training on the coefficients of the stochastic process, represented in the Fourier basis.

One specific use case is modelling changes in morphometry (i.e. shapes) of organisms in evolutionary biology. The morphometry of an organism can be modelled as points, curves or surfaces embedded in Euclidean space. Felsenstein [1985] suggests using Wiener processes to model changes in morphometry, for example, height. Sommer et al. [2021] propose extending this methodology to whole shapes by using stochastic flows to define diffeomorphisms on Euclidean space [Kunita, 1997]. Then, changes of the shapes are modelled deterministically as diffeomorphisms on $\mathbb{R}^d$ that act on the embeddings [Younes, 2019]. Recent work has generalised this to the stochastic setting by considering diffusion bridges between shapes [Arnaudon et al., 2019, 2022]. However, they first discretise the shapes and then find a diffusion bridge between the discretised shapes. In this work, we condition, and then we discretise. In doing so, we show that as the number of points goes to infinity, the bridge is still well-defined. Moreover, we may use other discretisations for shapes such as Fourier bases.

## 2 Related work and contributions

### 2.1 Related work

In finite dimensions, methods have been developed to approximate non-linear bridge processes of the form Equation (6). When conditioning on an end point $y$ at time $T$, the conditioned process contains an intractable term $\nabla_x \log p(t, x; T, y)$, called the score term. This term can be replaced with $\nabla_x \log \tilde{p}(t, x; T, y)$, where $\tilde{p}$ is the transition function from another SDE with a known closed form, such as Brownian motion or other linear processes [Delyon and Hu, 2006, van der Meulen and Schauer, 2022]. Then we can sample from the approximation instead, and use Monte Carlo methods using the ratio given by the Radon-Nikodym derivative as a likelihood ratio, thereby sampling from the true path distribution [van der Meulen and Schauer, 2022].

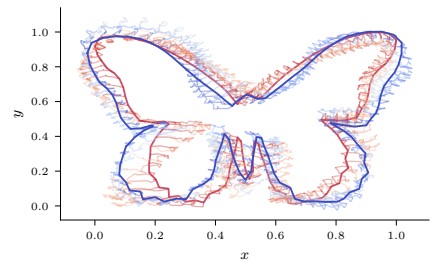

Figure 2: A stochastic process between two butterfly outlines (Papilio polytes in red, Parnassius honrathi in blue).

Recently, Heng et al. [2021] adapted the score-matching methods of Vincent [2011], Song and Ermon [2019], Song et al. [2021] to learn the score term for non-linear bridge processes. To do so, they introduce a new loss function to learn the time reversal of the process. They then learn the time reversal of the time reversal, which gives the forward bridge. Our work uses their method to learn the score term after discretising the SDE via truncated sums of basis elements. Phillips et al. [2022] also consider using truncated sums of basis elements for

discretising SDEs, however, only for infinite-dimensional Ornstein-Uhlenbeck processes, which are linear.

Recent work on generative modelling has investigated score matching for infinite-dimensional diffusion processes [Pidstrigach et al., 2023, Franzese et al., 2023, Bond-Taylor and Willcocks, 2023, Hagemann et al., 2023, Lim et al., 2023]. This problem is similar to our task of conditioning an SDE, but not the same: The main difference is that our SDEs are fixed, known a priori, and potentially nonlinear, whereas in generative modelling the SDE can be chosen freely. Hence, generative modelling often uses linear SDEs because the transition densities are known in closed form. In this sense, our problem relates to generative modelling but has a different setup.

In shape space, there is interest in defining stochastic bridges between shapes [Arnaudon et al., 2023]. Shapes are represented in the LDDMM framework [Younes, 2019], where they are modelled as embeddings in Euclidean space, e.g. curves and surfaces or sets of landmarks approximating a curve or surface. The deterministic image registration problem of matching two shapes is solved by finding a "minimum energy" mapping. More specifically, for two shapes $s_0, s_1 : \mathbb{R}^d \to \mathbb{R}^d$, we find a diffeomorphism $f : [0, T] \times \mathbb{R}^d \to \mathbb{R}^d$ such that $f(0, \cdot) = s_0(\cdot)$ and at $f(T, \cdot) = s_1(\cdot)$ and $f$ optimises a given energy functional [Bauer et al., 2014].

We employ a stochastic version of the LDDMM framework for our experiments. Instead of direct paths from $f_0$ to $f_T$ that optimise an energy functional, we define stochastic paths of diffeomorphisms between two shapes. For infinite-dimensional shapes such as curves, stochastic shape analysis was the focus of [Trouvé and Vialard, 2012, Vialard, 2013, Arnaudon et al., 2019], where stochastic processes are defined in the LDDMM framework. However, these do not explore the problem of conditioning the defined processes. In [Arnaudon et al., 2022], bridges between finite-dimensional shapes are derived. This is the first work to bridge between infinite-dimensional shapes.

### 2.2 Contributions

1. We derive Doob's $h$-transform for infinite-dimensional non-linear processes, allowing conditioning without first discretising the model.
2. We detail two models: one for direct conditioning on data and the second for assuming some observation error.
3. We use score matching to learn the score arising from the $h$-transform by training on the coefficients of the Fourier basis.
4. We demonstrate our method in modelling the changes in the shapes of butterflies over time.

## 3 Problem Statement

We assume that the data lives in a separable Hilbert space $(H, \langle \cdot, \cdot \rangle)$ and let $(\Omega, \mathcal{F}, \mathbb{P})$ be a probability space with natural filtration $\{\mathcal{F}_t\}$ and take $\mathcal{B}(H)$ to be the Borel algebra. We take a Wiener process $W$ on a separable Hilbert space $U$ (where $U$ can equal $H$) with covariance given by $Q$. We then consider Hilbert space-valued SDEs of the form

$$\mathrm{d}X(t) = (AX(t) + F(X(t)))\mathrm{d}t + B(X(t))\mathrm{d}W(t), \qquad X(0) = \xi_0 \in H, \qquad (1)$$

where $A : D(A) \subset H \to H$ is the infinitesimal generator of a strongly continuous semigroup, and $F : [0, T] \times H \to H$, $B : [0, T] \times H \to \mathrm{HS}(Q^{1/2}(U), H)$, where $\mathrm{HS}(Q^{1/2}(U), H)$ denotes the Hilbert-Schmidt operators from $Q^{1/2}(U)$ to $H$, and $Q^{1/2}$ is the unique, non-negative symmetric, linear operator on $U$ satisfying $Q^{1/2} \circ Q^{1/2} = Q$ (see Röckner and Claudia [2007, Proposition 2.3.4] for details). Note that $F$ and $B$ can depend non-linearly on $X(t)$, so other methods, such as Gaussian process regression, cannot be used to incorporate data. We also need the following assumptions about Equation (1):

**Assumption 3.1.** Equation (1) has a unique strong Markov solution denoted $X(t, \xi_0)$.

**Assumption 3.2.** The solution $X(t, \xi)$ is twice Fréchet differentiable with respect to the initial value $\xi$, with derivatives continuous on $[0, T] \times H$.

Assumption 3.1 is strong, but is satisfied when $A$ is bounded and $F$ and $B$ satisfy certain Lipschitz continuity and boundedness conditions. With some extra Fréchet differentiability conditions on $F$

and $B$, Assumption 3.2 will also be satisfied. See Da Prato and Zabczyk [2014, Part II] for details on solutions. Under the above setup, we consider two problems in conditioning the model on given observational data. First, we tackle the exact matching problem:

**Problem 3.1.** (Exact matching) Condition $X$ such that $X(T, \xi_0) \in \Gamma$ where $\Gamma \subseteq H$ is a set with nonzero measure.

More precisely, we define a new probability measure, under which the expectation equals the original expectation conditioned on the event that $X(T, \xi_0) \in \Gamma$: $\mathbb{E}_{new}[\cdot] = \mathbb{E}[\cdot \mid X(T, \xi_0) \in \Gamma]$. Section 5.2.1 discusses this. To solve this, we also require an extra assumption:

**Assumption 3.3.** The transition operator $P(\xi, t, \Gamma) := \mathbb{E}[\delta_\Gamma(X(t, \xi))]$ is twice Fréchet differentiable with respect to $\xi$, and once with respect to $t$ with continuous derivatives.

In infinite dimensions Assumption 3.3 is a strong assumption; however, we will study some specific sets $\Gamma$ for which this is satisfied in Sec. 5.3. Moreover, with extra conditions on $A$ and $Q$, there are also SDEs that satisfy this assumption for any nonzero measure Borel set. See Da Prato and Zabczyk [2014, Theorem 9.39 and Theorem 9.43] and Cerrai [2001, Section 6.5 and Section 7.3] for examples. The inexact matching problem does not require Assumption 3.3 and allows us to consider observation noise:

**Problem 3.2.** (Inexact matching) Condition $X$ so that $X(T, \xi_0)$ is "near" an observed function $V$.

Exact matching could be rephrased as conditioning such that at time $T$, the distance between $X(T, \xi_0)$ and a set $\Gamma$ is equal to 0. For the inexact matching, we instead condition such that the distance between $X(T, \xi_0)$ and a target set $\Gamma$ (or target function $V$) is Gaussian with mean 0. More generally, we can also take any differentiable radial basis function on the distance instead of a Gaussian.

For both problems, we show that the process $X(t, \xi_0)$ conditioned to exhibit the wanted behaviour at time $T$, will be a process $X^c(t, \xi_0)$ satisfying the SDE

$$\begin{cases} \mathrm{d}X^c(t) = [AX^c(t) + F(X^c(t))]\mathrm{d}t + B(X^c(t))\mathrm{d}W(t) \\ \qquad\qquad + B(X^c(t))B(X^c(t))^* \nabla \log h(t, X^c(t))\mathrm{d}t \\ X^c(0) = \xi_0, \end{cases} \tag{2}$$

where $\nabla_\xi \log(h(t, \xi))$ is a score function. For the exact matching problem, we will see that $h(t, \xi) = \mathbb{P}(X(T) \in \Gamma \mid X(t) = \xi) = \mathbb{E}[\delta_\Gamma(X(T - t, \xi))]$. For the inexact matching problem, we will instead set $h(t, \xi) = \mathbb{E}[\|f(X(T - t, \xi) - V\|_H; 0, \sigma)]$, for $f$ a Gaussian function, and $V$ a target function. The SDE in Equation (2) is analogous to the case of conditioning in finite dimensions, so the form may not be surprising. However, it is not obvious that this should work for non-linear equations in infinite dimensions.

## 4 Background

### 4.1 Strong Markov solutions

We consider Hilbert space-valued SDEs of the form Equation (1) for $W$ a $Q$-Wiener process in a Hilbert space $U$, where $Q$ can be the identity operator. We only consider strong solutions to Equation (1). An $H$-valued predictable process $X$ is a strong solution of Equation (1) if $\forall t \in [0, T]$ it satisfies a well-defined integral

$$X(t) = \xi + \int_0^t [AX(s) + F(s, X(s))]\mathrm{d}s + \int_0^t B(s, X(s))\mathrm{d}W(s). \tag{3}$$

We refer to Da Prato and Zabczyk [2014] for a discussion on the existence of strong solutions (and more general solutions). However, when $F, B$ satisfy some Lipschitz and linear growth conditions, and when $A$ is bounded, Equation (1) has a unique strong solution. Since this is true for any initial value $\xi$, we use the notation $X(t, \xi)$ to mean the unique solution of Equation (1) with initial value $\xi$. This solution is a Markov process and for $f : H \to \mathbb{R}$ a bounded function, measurable with respect to the Borel algebra the transition operators $P_t f(\xi) = \mathbb{E}[f(X(t, \xi))]$ satisfy the Markov condition:

$$\mathbb{E}[\psi(X(t, \xi)) \mid \mathcal{F}_s] = \mathbb{E}[\psi(X(t - s, X(s, \xi)))] = \mathbb{E}[\psi(X(t, \xi)) \mid X(s, \xi)]. \tag{4}$$

This says that the expected value of the solution at a time $t$ from a starting value $\xi$, given all the information from some previous time $s$, is the same as the expected value of the solution started at value $X(s, \xi)$ at time $t - s$.

## 4.2 Doob's $h$-transform in finite dimensions

In order to give more intuition for the infinite-dimensional Doob's $h$-transform, we present an informal introduction to the topic. This is all well-known, and the details can be found, for example, in Rogers and Williams [2000, Chapter 6]. Doob's $h$-transform in finite dimensions is a useful theory for conditioning stochastic differential equations. For example, suppose we have an SDE in $\mathbb{R}^d$

$$x(t) = f(t, x(t))\mathrm{d}t + \sigma(t, x(t))\mathrm{d}W(t), \quad x(0) = x_0 \in \mathbb{R}^d \tag{5}$$

and we want to condition this SDE to hit $y$ at time $T$. This corresponds to finding a measure $\mathbb{Q}$ such that $\mathbb{E}^{\mathbb{Q}}[x(t)] = \mathbb{E}[x(t) \mid x(T) = y]$. Doob's $h$-transform allows us to define such a measure using so called $h$-functions. Let $p(t, y; t + s, y')$ be the transition density of $x_t$, defined as $\mathbb{E}[x(t + s) \in A \mid x(t) = y] = \int_A p(t, y; t + s, y')\mathrm{d}y'$. Let $h : [0, T] \times \mathbb{R}^d \to (0, \infty)$ be a function satisfying $h(t, x) = \int h(t + s, y)p(t, x; t + s, y)\mathrm{d}y$, such that for $z(t) := h(t, x(t))$, $\mathbb{E}[z(T)] = 1$. Then, $z(t)$ is a martingale and there exists a measure $\mathbb{Q}$ such that $\frac{\mathrm{d}\mathbb{Q}}{\mathrm{d}\mathbb{P}}|_{\mathcal{F}_t} = z_t$. Moreover, under this measure $\mathbb{Q}$, $x(t)$ satisfies a new SDE

$$\mathrm{d}x^c(t) = f(t, x^c(t))\mathrm{d}t + \sigma\sigma^\top(t, x^c)\nabla_x \log h(t, x^c(t))\mathrm{d}t + \sigma(t, x^c(t))\mathrm{d}W(t), \quad x^c(0) = x_0. \tag{6}$$

The SDE in Equation (6) can be thought of as a conditioned version of the original SDE in Equation (5). For example, consider what happens when we take $h(t, x) := \frac{p(t, x; T, y)}{p(0, x_0; T, y)}$. Then for a function $f$

$$\mathbb{E}^{\mathbb{Q}}[f(x(t))] = \int f(z)\frac{p(t, z; T, y)}{p(0, x_0; T, y)}p(0, x_0; t, z)\mathrm{d}z = \mathbb{E}[f(x(t)) \mid x(T) = y]. \tag{7}$$

This is one example of Doob's $h$-transform but other $h$ functions can also be defined, for example, to condition $x_t$ to stay within certain bounds, or not to go above a certain value for a certain time period.

For $h(t, x) := \frac{p(t, x; T, y)}{p(0, x_0; T, y)}$, as in conditioning on an end point, there is, in general, no closed form solution for $h$. Different methods to learn the bridge exist [Delyon and Hu, 2006, Schauer et al., 2017]. More recently, score-based learning methods were proposed to learn the term $\nabla_x \log p(t, x; T, y)$ [Heng et al., 2021], which we will adapt to the infinite-dimensional setting.

## 5 Method

We are interested in conditioning the stochastic process to exhibit a particular behaviour at the end time $T$. We consider two scenarios. The first is exact matching: we condition such that given a set $\Gamma \subset H$, then $X(T, \xi_0) \in \Gamma$. The second is inexact matching: for some $Y \in H$ we condition such that as $t$ approaches $T$, $X(t, \xi_0)$ becomes "close" to $Y$.

We proceed as follows. First, we show that we can define a new probability measure given an appropriate random variable. When we have shown that this is possible, we will discuss options for the random variable. We will give specific variables, show that they fit some necessary conditions, and solve Problem 3.1 and Problem 3.2.

### 5.1 Doob's $h$-transform in infinite dimensions

Here, we suppose that we already have an appropriate function $h$ which we show that we can use to rescale our original probability distribution, giving us a conditioned probability.

**Theorem 5.1.** *Let $h : [0, T] \times H \to \mathbb{R}_{>0}$ be a continuous function twice Fréchet differentiable with respect to $\xi \in H$ and once differentiable with respect to $t$, with continuous derivatives. Suppose $X$ is the strong solution to the stochastic differential equation in Equation* (1). *Moreover, we assume that $Z(t) := h(t, X(t))$ is a strictly positive martingale, with $Z(0) = 1$, and $\mathbb{E}[Z(T)] = 1$.*

*Then $\mathrm{d}\widehat{\mathbb{P}} := Z(T)\mathrm{d}\mathbb{P}$ defines a new probability measure. Moreover, $X$ satisfies the SDE*

$$\begin{aligned} X(t) =& X(0) + \int_0^t B(X(s))B(X(s))^* \nabla \log h(s, X(s))\mathrm{d}s \\ &+ \int_0^t [AX(s) + F(X(s))]\mathrm{d}s + \int_0^t B(X(s))\mathrm{d}\widehat{W}(s), \end{aligned} \tag{8}$$

*where $\widehat{W}$ is the Wiener process with respect to the measure $\widehat{\mathbb{P}}$.*

*Proof.* First, we show that $Z(t) \coloneqq h(t, X(t))$ defines a continuous martingale and apply an infinite-dimensional Itô's theorem followed by Doléans exponential to rewrite $Z$. We then may apply the infinite-dimensional Girsanov's theorem, and rewrite the original SDE in terms of the resulting Wiener process $\widehat{W}$. See Theorem C.1 for full details. □

## 5.2 Defining the transforms

Previously, we showed that given a function $h$ satisfying certain conditions, we can use this to weight the probability measure, giving us a new conditioned probability measure. Now, we address which $h$ functions to use and the properties of the resulting processes. In infinite dimensions, there is no measure that satisfies all properties of the usual finite-dimensional Lebesgue measure and so it does not make sense to consider transition densities. However, transition operators of the form $P_t f(\xi) = \mathbb{E}[f(X(t, \xi))]$ exist and satisfy the Markov property in Equation (4), so we opt to use these instead. For a strictly positive, bounded Borel function $\psi : H \to \mathbb{R}$, we take functions of the form

$$h(t, \xi) = C_T \cdot \mathbb{E}[\psi(X(T - t, \xi))]. \tag{9}$$

where $C_T = 1/\mathbb{E}[\psi(X(T, \xi_0)]$ is a normalising constant. Consider, for example, when the function $\psi$ is the Dirac delta function of some set $A$ with non-zero measure. Then $h(t, \xi)$ is the probability that at the end time, the process will be in the set $\Gamma$, given that at the current time, the process is equal to $\xi$. Functions of the form Equation (9) satisfy some of the necessary assumptions on $h$ for Theorem 5.1: they are positive martingales, with $Z(0) = 1$ and $\mathbb{E}[Z(T)] = 1$.

**Lemma 5.2.** *Let $X$ be as in Equation* (1)*, satisfying Assumption 3.1. Given a function $h : [0, T] \times H \to \mathbb{R}$ satisfying Equation* (9)*, $Z(t) \coloneqq h(t, X(t, \xi_0))$ is a strictly positive martingale such that $Z(0) = 1$ and $Z(T) = C_T \cdot \psi(X(T, \xi_0))$.*

*Proof.* Apply the tower property and check the assertions hold. See Lemma C.3 for full details. □

The other necessary condition on $h$ is differentiability. For this, we consider the specific functions with different $\psi$'s separately for the exact and inexact matching cases.

### 5.2.1 Exact matching

Let $\Gamma \in H$ be Borel measurable, and suppose that Assumption 3.3 holds. For some instances where this holds, see Da Prato and Zabczyk [2014, Chapter 9], or Sec. 5.3. Then we define

$$h(t, \xi) \coloneqq \mathbb{E}[\delta_\Gamma(X(T - t, \xi))]/\mathbb{E}[\delta_\Gamma(X(T, \xi_0))]. \tag{10}$$

The proof that this $h$-transform does solve Problem 3.1 is similar to the finite-dimensional version. Note that $h(T, X(T)) = \delta_\Gamma(X(T, \xi_0))/\mathbb{E}[\delta_\Gamma(X(T, \xi_0))]$. Hence, for some random variable $Y$ we see:

$$\widehat{\mathbb{E}}[Y] = \int_\Omega Y \mathrm{d}\widehat{\mathbb{P}} = \mathbb{P}(\{X(T, \xi_0) \in \Gamma\})^{-1} \int_{\{X(T) \in \Gamma\}} Y \mathrm{d}\mathbb{P} = \mathbb{E}[Y \mid X(T, \xi_0) \in \Gamma]. \tag{11}$$

### 5.2.2 Inexact matching

In addition to the exact matching problem, we can solve the inexact matching problem by conditioning the process such that, at the end time, it approximates some behaviour. This has two advantages. Firstly, we do not need Assumption 3.3 and instead use Assumption 3.2. Assumption 3.2 is satisfied when $F$ and $B$ satisfy Fréchet differentiable conditions. The second advantage is that this allows us to account for observation noise in models. We condition on the Gaussian distance between the endpoint and some target point or observation by defining a function $\psi : H \to \mathbb{R}$ that is twice Fréchet differentiable. Then under Assumptions 3.1 and 3.2 i.e. $X(t)$ is a strong solution, a Markov process and is twice differentiable with respect to the initial value, the function $h(t, \xi) \coloneqq \mathbb{E}[\psi(X(T - t, \xi))]$ will satisfy the necessary conditions given at the start of Section 5.1:

**Lemma 5.3.** *Let $\psi : H \to \mathbb{R}$ be a continuous function, twice Fréchet differentiable, with continuous derivatives. Then $h(t, \xi) \coloneqq \mathbb{E}[\psi(X(T - t, \xi))]$ is twice Fréchet differentiable in $\xi$ and once differentiable in $t$, with continuous derivatives.*

Apply Itô's formula to $h$ and use properties of expectation to differentiate. See Lemma C.4 for the details.

One such function satisfying Lemma 5.3 is the Gaussian kernel function $k : H \times H \to \mathbb{R}$,

$$k_\sigma(V, X) = \frac{1}{\sqrt{2\pi\sigma}} \exp\left(-\frac{1}{2\sigma}\|X - V\|_H^2\right). \tag{12}$$

Here, we fix an observation $V \in H$ and parameter $\sigma \in \mathbb{R}$ and vary $X \in H$. The function $k$ is twice Fréchet differentiable in each argument, with continuous derivatives; hence the function $h(t, \xi) = \mathbb{E}[k_\sigma(V, X(t, \xi))]$ satisfies the requirements of Lemma 5.3. Moreover, this gives us a method of including observation noise in our model. In finite dimensions, to model inexact matching for a stochastic process $x(t) \in \mathbb{R}^d$, one can take the function

$$h(t, x) := \int f_d(v; y, \Sigma) p(t, x; T, y) \mathrm{d}y, \tag{13}$$

where $v \in \mathbb{R}^d$ is a target value, $p(t, x; T, y)$ is the transition density of the $\mathbb{R}^d$-valued stochastic process and $f_d(\cdot; \mu, \Sigma)$ is the density of the normal distribution on $\mathbb{R}^d$ with mean $\mu \in \mathbb{R}^d$ and covariance $\Sigma \in \mathbb{R}^{d \times d}$. See Arnaudon et al. [2022, Section 3.1] for more details. To compare, for $f_1(\cdot; 0, \sigma)$ the density of the one-dimensional normal distribution with mean 0 and variance $\sigma \in \mathbb{R}$, and $P(T - t, \xi, \Gamma) = \mathbb{P}[X(T) \in \Gamma \mid X(t) = \xi]$, we set

$$h(t, \xi) := \mathbb{E}[k_\sigma(V, X(t, \xi))] = \int_H f(\|\gamma - V\|_H^2; 0, \sigma) P(T - t, \xi, \mathrm{d}\gamma). \tag{14}$$

Defining the function in this way means we condition on a distance between $X(T)$ and our observation. It also means we can change the distance function to another similarity measure. For example, when the functions represent shapes, we could use another norm that measures the dissimilarity of shapes as in Pennec et al. [2020, Chapter 12].

## 5.3 Sampling from infinite-dimensional $h$-transforms

Thus far, we have shown that in infinite dimensions, we can condition stochastic processes either exactly or inexactly, and the conditioned process has form Equation (2). We now turn our attention to sampling from these conditioned processes. For this we discuss how to discretise.

For an orthonormal basis $\{e_i\}_{i=1}^\infty$ of a separable Hilbert space $H$, let $H^N = \mathrm{span}(\{e_i\}_{i=1}^N) \subset H$. Let $X(t, \xi)$ be a strong solution to Equation (1), with $X(0) = \xi$. Since strong solutions are also weak solutions [Da Prato and Zabczyk, 2014, Chapter 6.1], we can write $X(t, \xi)$ as a sum of finite dimensional SDEs, with each finite SDE satisfying

$$\langle X(t), e_i \rangle = \langle \xi, e_i \rangle + \int_0^t \langle AX(s) + f(X(s)), e_i \rangle \mathrm{d}s + \int_0^t \langle e_i, B(X(s)) \mathrm{d}W(s) \rangle. \tag{15}$$

Using Equation (15), we can define an SDE $X^N$ as $X_i^N(t) := \langle X(t), e_i \rangle$, where $X_i^N$ is the $i^{\text{th}}$ component of $X^N \in \mathbb{R}^N$. For finite dimensional sets $\Gamma_i \subset \mathbb{R}$ we look at the problem of conditioning on cylindrical sets of the form

$$\Gamma_N = \{\varphi \in H \mid \varphi_i \in \Gamma_i, \, 1 \leq i \leq N\}, \tag{16}$$

for $\phi_i = \langle \phi, e_i \rangle$.

**Lemma 5.4.** *Let* $\Gamma_N$ *be as in Equation* (16) *and* $h : [0, T] \times H^N \to \mathbb{R}$ *be defined by* $h(t, Y) := \mathbb{E}[\delta_{\Gamma_N}(X(T - t, Y))]$. *Moreover, define* $g : [0, T] \times \mathbb{R}^N \to \mathbb{R}$ *by* $g(t, y) := \mathbb{E}[\prod_{i=1}^N \delta_{\Gamma_i}(X_i^N(T - t, y))]$. *Then* $\langle \nabla \log h(t, Y), e_i \rangle = [\nabla \log g(t, (Y_i)_{i=1}^N)]_i$.

*Proof.* It follows by noting that $\mathbb{E}[\delta_{\Gamma_N}(Y)] = \mathbb{E}[\prod_{i=1}^N \delta_{\Gamma_i}(Y_i)]$. See Lemma C.5 for details. $\square$

Since $h(t, Y) = g(t, (Y_i)_{i=1}^N)$, the sets $\Gamma_N$ satisfy Assumption 3.3 as long as the sets $\Gamma_i$ do in finite dimensions. We have shown that conditioning on sets only depending on the first $N$ eigenvalues is equivalent to conditioning the $N$ dimensional projection of the SDE onto the first $N$ basis elements.

Table 1: A comparison of the trained score of the Brownian motion process.

| | Fourier (num. bases) | | | Landmarks (num. pts) | | |
|---|---|---|---|---|---|---|
| | 8 | 16 | 32 | 8 | 16 | 32 |
| RMSE | 5.09 | 6.66 | 10.54 | 7.95 | 6.08 | 10.79 |
| Time (s) | 105.1 | 201.8 | 949.4 | 95.9 | 104.8 | 183.0 |
| Epochs | 100 | 150 | 300 | 100 | 100 | 100 |

With this discretisation onto finite dimensions, we can adapt a finite-dimensional algorithm to sample from the finite-dimensional bridges. For this we opt for using the algorithm in Heng et al. [2021], since it can be easily applied to Problems 3.1 and 3.2 and we can scale up to higher dimensions by using a different network architecture. Here, they leverage the diffusion approximation (in this case, Euler-Maruyama) and score-matching techniques to first learn the time reversal of the diffusion process. Applying the algorithm again on the time reversal, started at the proposed end point, gives the forward-in-time diffusion bridge.

# 6 Experiments

We consider two main setups. Firstly we look at Brownian motion between shapes and use this to evaluate our method, since for Brownian motion we have a closed form solution for the score function. We then apply this to problems from the shape space literature. There, they are interested in stochastic bridges between shapes which has applications within medical imaging and evolutionary biology [Gerig et al., 2001, Arnaudon et al., 2017, 2023]. We expand on that body of work, by allowing shapes to be treated as infinite-dimensional objects when bridging, as in the non-stochastic case [Younes, 2019]. Until now, this was impossible for stochastic shape paths, since the theory for this was missing.

The code used for our training and experiments can be found at `https://github.com/libbylbaker/infsdebridge` and further details on experiments can be found in Appendix B.

## 6.1 Brownian Motion

For Brownian motion between shapes, we look at using both discretisations via landmarks and Fourier bases, for conditioning both exactly and inexactly and compare to the true solution. We train on a target shape of a circle with radius 1. For the landmark discretisation, we condition such that the landmarks of the process end at the landmarks of the target shape, and for the Fourier basis we condition such that the chosen basis elements are equal to the Fourier basis of the circle at the end time. In Tab. 1 we give the mean square error for different numbers of dimensions. For the Fourier basis, we evaluate the score on 100 points, and find the error between this and the true value so that we may compare to the landmark errors. We see that as the dimensions grow, we need a larger training time to maintain lower errors. We note that each Fourier basis, contains two parts: the real and imaginary. We train on batches of 50 SDE trajectories, with 40 batches per epoch. For training details see Appendix B.1.

## 6.2 Experiments on shape space

Next we turn to a concrete problem: we model the change in morphometry (shapes) of butterflies over time. Studying changes of morphometry of organisms over time is important to evolutionary biologists. For example, for butterflies, one can ask whether the change in wing shape correlates with a change in habitat or climate. Rather than extracting finite-dimensional information from the shapes, such as height or a subset of chosen points, we apply the analysis to the entire shape, as suggested in Sommer et al. [2021]. Being able to condition between shapes is a key step in phylogenetic inference, where it will be applied to compute likelihoods of phylogenetic trees from morphological data. Until now, it was only possible for finite-dimensional extractions of the shape. Future work will consider extending our methods for parameter estimation of SDEs for phylogenetic inference. This is in order to extend the Brownian motion model of trait evolution to shapes where the SDE models the transitions over the edges in the phylogenetic tree [Felsenstein, 1985].

### 6.2.1 SDEs in shape space

For SDEs in shape space we take the SDE defined in Sommer et al. [2021] as

$$dX(t) = \int_0^t Q(X(t))dW(t) \qquad (Qh)f(x) = \int_{\mathbb{R}^2} k(h(x), y)f(y)dy. \tag{17}$$

where for each $h \in H = L^2(\mathbb{R}^2, \mathbb{R}^2)$, $Q(h) : H \to H$ is a Hilbert-Schmidt operator, and $k$ is a smooth kernel $k \in L^2(\mathbb{R}^2 \times \mathbb{R}^2, \mathbb{R}^2)$.

This corresponds to a stochastic flow of diffeomorphisms on $\mathbb{R}^2$, with a Brownian temporal model. For each $x \in \mathbb{R}^2$, $X(t, x)$ models the position of $x \in \mathbb{R}^2$ at time $t$, and the function $x \to X(t, x)$ is a diffeomorphism for all $t$. To see this we write this in the language of stochastic flows as defined in Kunita [1997]. Define the martingale $F(t, x) := QW(t, x)$, where $W$ is the Wiener process on $H$ and $Q$ is defined as before. Then, we define the stochastic flow of the martingale $F$ as

$$p(t, x) = x + \int_0^t F(p(r, x), dr), \tag{18}$$

where the integral is defined in Kunita [1997, Chapter 3]. Then by Kunita [1997, Theorem 4.6.5] if $k$ is smooth, the map $p(t, \cdot)$ is a diffeomorphism for each $t$. See also Da Prato and Zabczyk [2014, Chapter 0.1] for general details of lifts of diffusion processes to infinite dimensions.

In Figure 8 we plot some example trajectories of Equation (17) for various parameters, with a circle as the initial value. In Figure 6 we plot one trajectory for a butterfly. The trajectories are calculated in terms of a Fourier basis and for Figure 6 we plot the trajectories of a subset of points of the shape where we see the temporal Brownian model.

### 6.2.2 Results

We illustrate our method on butterfly data. To demonstrate, we first use two butterflies with somewhat different shapes [GBIF.Org User, 2024]. One trajectory between the two butterflies is plotted in Figure 2. In this, we can see the high correlation between neighbouring points, with a Brownian temporal model. In Figure 1, we plot 120 butterfly trajectories, at specific time points. For $t = 0.2$ we see that the butterfly outlines are mostly close to the start butterfly in pink, and at time $t = 0.8$, they are closer to the green target butterfly. In Appendix B, we also plot the score function over time for varying numbers of basis elements Figure 3.

For the next experiment, we take fifty butterfly specimens across five closely related species from the Papilio family (see Figure 5) [Kawahara et al., 2023]. The butterflies are aligned via Procrustes, and a mean consensus shape is obtained using Geomorph [Adams and Otarola-Castillo, 2013]. More details of the butterflies and their processing are in Appendix A. We train our model on the mean of the butterflies to learn the time reversal from any given input, to hit the distribution at time $T = 1.0$, which we plot in Figure 4.

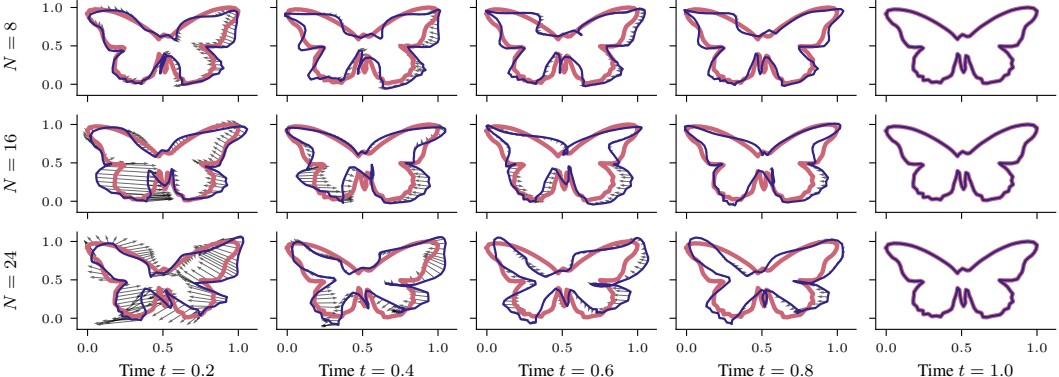

Figure 3: Score fields evaluated on a selection of points at different time steps. In general, the score field is expected to "push" the shape towards the target. We show the learned score fields (black arrows) represented by varying numbers $N$ of base functions at different time steps, as well as the current shape (blue curves) and the target shape (red curves).

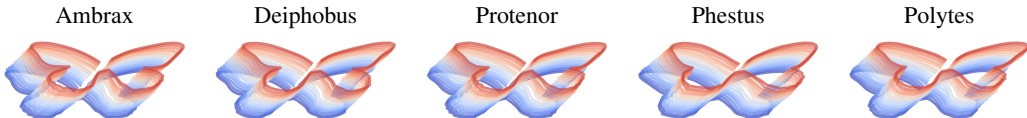

| Ambrax | Deiphobus | Protenor | Phestus | Polytes |

Figure 4: We use a dataset of 40 closely related butterflies with five different species. We find a mean across the dataset and plot single trajectories between the mean at time $t = 0$ (in blue) and a specimen from each species at time $t = 1$ (in red).

## 7    Conclusion, limitations and future work

We have proved that Doob's $h$-transform can also be used in infinite-dimensions for stochastic differential equations with strong solutions. We can condition non-linear function-valued stochastic models on observations, either directly on data or by including observation noise. The conditioned stochastic process satisfies a new differential equation involving a score function, which we can approximate using score learning. However, due to the reliance on Itô's formula, it would be hard to generalise this proof to non-strong solutions.

To learn the score, we used the architecture detailed in Figure 7. Although this seemed to work well for our experiments, more research could go into the network architecture which could further increase the dimensions that we consider. Furthermore, we only do the first step in Heng et al. [2021] and learn the time reversal since error compounds in learning the forward bridge. Future work will consider how well the time reversal approximates the forward bridge or how to learn the forward bridge directly. Moreover, as previously mentioned, learning Doob's $h$-transform is only the first step in phylogenetic inference for shapes of species. Future work will consider how to expand the infinite-dimensional bridges to inference problems.

## Acknowledgments and Disclosure of Funding

The work presented in this article was done at the Center for Computational Evolutionary Morphometry and is partly supported by Novo Nordisk Foundation grant NNF18OC0052000, a research grant (VIL40582) from VILLUM FONDEN, and UCPH Data+ Strategy 2023 funds for interdisciplinary research.

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

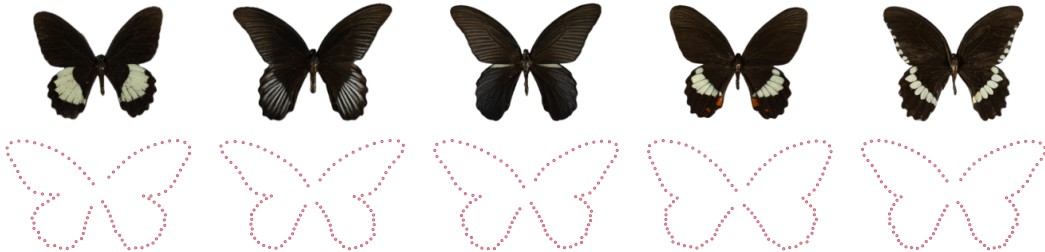

Figure 5: The five closely related species of *Papilio*, from left to right; *Papilio Ambrax*, *Papilio Deiphobus*, *Papilio Protenor*, *Papilio Phestus* and *Papilio Polytes*. A subset of the landmarks for each specimen is shown underneath each corresponding image.

Olaf Ronneberger, Philipp Fischer, and Thomas Brox. U-net: Convolutional networks for biomedical image segmentation. In *Medical Image Computing and Computer-Assisted Intervention–MICCAI 2015: 18th International Conference, Munich, Germany, October 5-9, 2015, Proceedings, Part III 18*, pages 234–241. Springer, 2015.

Ashish Vaswani, Noam Shazeer, Niki Parmar, Jakob Uszkoreit, Llion Jones, Aidan N Gomez, Łukasz Kaiser, and Illia Polosukhin. Attention is all you need. *Advances in neural information processing systems*, 30, 2017.

Zdzislaw Brzezniak, Jan van Neerven, Mark Veraar, and Lutz Weis. Itô's formula in UMD Banach spaces and regularity of solution of the Zakai equation. *Journal of Differential Equations, 245, no. 1, p. 30-58, 2008*, 245, 07 2008. doi: 10.1016/j.jde.2008.03.026.

Giuseppe Da Prato, Arnulf Jentzen, and Michael Roeckner. A mild Itô formula for SPDEs. *Transactions of the American Mathematical Society*, 372(6):3755–3807, June 2019. ISSN 0002-9947, 1088-6850. doi: 10.1090/tran/7165.

## A Butterfly Processing

The Lepidoptera images originate from five closely related species within the genus *Papilio* from the Papilionidae family [Kawahara et al., 2023]. The images are obtained through gbif.org [gbif.org, 2023], filtering within *preserved material* from museum collections. The images are segmented with the Python packages *Segment Anything* [Kirillov et al., 2023] and *Grounding Dino* [Liu et al., 2023]. The thorax's contour is removed from the outline by localising the horizontal local minimum in the outline on both the top and bottom sides of the thorax, corresponding to four anatomical landmarks where the wing is mounted to the thorax. The separation landmark of the fore and hind wings is set by identifying the vertical valley of the outline on both the left and right sides. The landmarks are used to place 250 evenly spaced semi-landmarks by interpolating the segmentation outline for each wing—images with incorrectly placed landmarks, specimens with broken wings, or abnormalities are manually removed during the process.

Eight random images were drawn for each of the five species. In total, 40 sets of 1000 landmarks were aligned using Procrustes alignment. The alignment and the mean consensus shape were obtained by using the R package *Geomorph v. 4.06* [Adams and Otarola-Castillo, 2013]. See Figure 5 for examples of the five species of butterflies and a subsample of the aligned landmarks.

## B Experiment details and further figures

### B.1 Score Learning

Given an SDE discretised over the first $N$ coefficients of a basis function, we learn the score function. To do this, we use the algorithm presented in Heng et al. [2021] for finding the score function of a finite-dimensional Markov process $x(t)$ with transition function $p(x(t) \mid x(0)), 0 \leq t \leq T$, given by $\nabla \log p(x(T) \mid x(t))$. Here, they leverage the diffusion approximation (in this case, Euler-Maruyama) and score-matching techniques to

Table 2: Training configurations for learning scores with different numbers of bases

| Num. bases | Input/output dims | Time embedding dims | Downsampling dims | Upsampling dims | Activation |
|---|---|---|---|---|---|
| 8 | 32 | 32 | [64, 32, 16, 8] | [8, 16, 32, 64] | silu |
| 16 | 64 | 64 | [128, 64, 32, 16] | [16, 32, 64, 128] | silu |
| 32 | 128 | 128 | [256, 128, 64, 32] | [32, 64, 128, 256] | silu |

first learn the time reversal of the diffusion process. Applying the algorithm again on the time reversal, started at the proposed end point, gives the forward-in-time diffusion bridge.

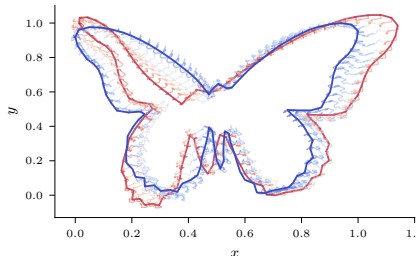

Figure 6: Sample from the SDE of Equation (17).

We found that the neural network architecture outlined in Heng et al. [2021] did not scale well to learning higher dimensional SDEs. We therefore used a different structure (see Figure 7). We use a U-net [Ronneberger et al., 2015] structure with skip connections in the form of fully connected layers to scale up the network's capacity. The time step information is encoded by the well-known sinusoidal embedding proposed in Vaswani et al. [2017] and added element-wise to the outputs of the fully connected layers. Finally, the real denoising score matching loss function proposed in Heng et al. [2021] is computed and the stochastic gradient descent is used to update the network parameters.

The exact specification we used for different bases or points is given in Table 2. We used the Adam optimiser for the training, with a starting learning rate of 0.0001 and 500 warmup steps. After warming up, the learning rate decreases cosinely until it reaches 1e-6. All the training and evaluation computations were done with one NVIDIA RTX 4090 GPU and one Intel(R) Xeon(R) CPU E5-2650 v4 @ 2.20GHz.

## B.2 Shape Spaces

### B.2.1 Discretisation

We look at both discretisations in terms of the Fourier basis, and in terms of points. For points we take the discretisation

$$\mathrm{d}x_i(t) = x_{0,i} + \sum_{y \in \mathcal{G}} k(x_i(t), y)\delta(y)\mathrm{d}w_y(t), \tag{19}$$

where $\mathcal{G}$ is a set of grid point in $\mathbb{R}^2$, and $w_y(t) \in \mathbb{R}^2$ a Wiener process. For the Fourier basis, writing the SDE in Equation (17) into basis elements, gives

$$\mathrm{d}X(t) = \sum_{n=1}^{\infty} \sum_{l,m=1}^{\infty} \langle e_n, Q(X(t))(g_{l,m})\rangle \mathrm{d}w_{l,m}(t)e_n, \tag{20}$$

where $e_n(x) = e^{inx}$ and $g_{l,m}(x_1, x_2) = e^{ilx_1 + imx_2}$. Then, we truncate the bases to $n \leq N$ and $l, m \leq M$ elements. The values in the sum can be approximated as follows:

$$\langle e_n, \tilde{Q}(X(t))(g_{l,m})(x) \rangle \approx \frac{1}{2\pi} \sum_{x \in \mathcal{G}_1} e^{inx} \sum_{y \in \mathcal{G}_2} k(X(t)(x), y)g_{l,m}(y)\Delta y \Delta x, \tag{21}$$

where $\mathcal{G}_1$ and $\mathcal{G}_2$ are grids over $[-\pi, \pi]$ and $D \subset \mathbb{R}^2$. The inner sum can be computed as a fast Fourier transform, and the outer as a 2-dimensional, inverse fast Fourier transform. For the function $k$, we use the Gaussian kernel, with varying values for the covariance. In Appendix B, we apply this SDE, with various parameters, to a circle embedded into $\mathbb{R}^2$ in Figure 8, and to a butterfly in Figure 6.

### B.2.2 Further experiments

In Figure 8, we apply the unconditioned SDE in 20 to a circle, embedded into $\mathbb{R}^2$. The parameter $\sigma$ is the variance of the kernel $k$. We see that for increasing values of $\sigma$, the process becomes smoother. This makes sense, since for each point $y \in \mathbb{R}^2$, we can associate a noise field. Larger values of $\sigma$ for the kernels, mean the noise fields are wider and therefore points are more highly correlated leading to smoother shapes.

We see that increasing the number of basis elements used, initially leads to slightly noisier shapes which is to be expected, since higher basis elements contain the higher frequencies and details. However, the process seems to converge quite quickly, and there appears very little difference between $N = 16$ and $N = 24$ basis elements.

In Figure 6, we show the process started on a butterfly, with $\sigma = 0.1$ and eight landmarks, where the evolution starts from a fixed butterfly shape (in blue) and continues until time $t = 1$ (in red).

Figure 7: The neural network structure for approximating the discretised score function. A U-net architecture with skip connections (dashed lines) is used. Each layer consists of two dense layers activated by SiLU functions. Batch normalisation is applied to the end of layer (not shown). The time step $t$ is encoded using the sinusoidal embedding and added element-wise to the outputs of dense layers.

## C   Proofs

**Theorem C.1.** *(Theorem 5.1 in paper) Let $h : [0, T] \times H \to \mathbb{R}_{>0}$ be a continuous function twice Fréchet differentiable with respect to $\xi \in H$ and once differentiable with respect to $t$, with continuous derivatives. Suppose $X$ is the strong solution to the stochastic differential equation in Equation* (1). *Moreover, we assume that $Z(t) := h(t, X(t))$ is a strictly positive martingale, with $Z(0) = 1$, and $\mathbb{E}[Z(T)] = 1$.*

Unconditional Forward Trajectories

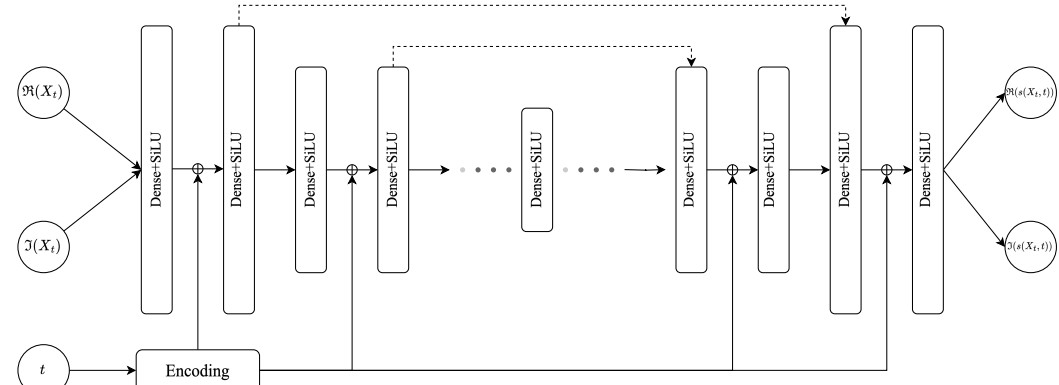

Figure 8: We visualise the effect of the SDE on a circle, for varying covariance of the Gaussian kernel $\sigma$, and different numbers of basis elements $N$.

Then $\mathrm{d}\widehat{\mathbb{P}} := Z(T)\mathrm{d}\mathbb{P}$ *defines a new probability measure. Moreover, $X$ satisfies the SDE*

$$\begin{aligned} X(t) =& X(0) + \int_0^t B(X(s))B(X(s))^*\nabla\log h(s, X(s))\mathrm{d}s \\ &+ \int_0^t [AX(s) + F(X(s))]\mathrm{d}s + \int_0^t B(X(s))\mathrm{d}\widehat{W}(s), \end{aligned} \tag{22}$$

*where $\widehat{W}$ is the Wiener process with respect to the measure $\widehat{\mathbb{P}}$.*

We split the proof into two and start with a lemma showing that $Z(t) := h(t, X(t))$ can be written in terms of an exponential.

**Lemma C.2.** *Let $h : [0, T] \times H \to \mathbb{R}$ and $Z(t) := h(t, X(t))$ satisfy the assumptions of Theorem 5.1. Then $Z(t) = \exp\left(L(t) - \frac{[L](t)}{2}\right)$, where*

$$L(t) = \int_0^t \langle B(X(s))^*\nabla\log h(s, X(s)), \mathrm{d}W(s)\rangle_{Q^{1/2}(U)}, \tag{23}$$

*and $[L]$ is the quadratic variation of L.*

*Proof.* We denote the $j^{\text{th}}$ Fréchet derivative with respect to the $i^{\text{th}}$ argument of a function $f$ by $D_i^j f$. In case $j = 1$, we will simply write $D_i f$. First, we apply the infinite-dimensional Itô's lemma included in Theorem D.2 [Brzezniak et al., 2008]. We can do this since we assume that $h$ and its Fréchet derivatives $D_1 h, D_2 h, D_2^2 h$ exist and are continuous. Moreover, $X$ is a strong solution, so $B(X(s))$ is stochastically integrable, and $AX(s) + F(X(s))$ is integrable and adapted. Furthermore, since we assume that $h(t, X(t))$ is a martingale, the drift terms arising in Itô's lemma must disappear. Therefore for $Z(t) := h(t, X(t))$,

$$Z(t) = h(0, X(0)) + \int_0^t D_2 h(s, X(s))B(X(s))\mathrm{d}W(s). \tag{24}$$

Now, we write $Z$ in the form of an exponential. For this, we use Doléans exponential. Set

$$L(t) = \log Z(0) + \int \frac{1}{Z_s}\mathrm{d}Z_s. \tag{25}$$

By assumption $Z$ is continuous and strictly positive, so via the Doléans-Dade exponential [Rogers and Williams, 2000, Chapter 3], it holds that

$$Z(t) = \exp\left(L(t) - \frac{[L](t)}{2}\right). \tag{26}$$

Since we defined $Z(t) = h(t, X(t))$, and we assume that $Z(0) = 1$, we know that

$$L(t) = \int_0^t D_2 \log h(s, X(s))B(X(s))\mathrm{d}W(s). \tag{27}$$

By the Riesz representation theorem, there exists an element in $H$ that we denote $\nabla\log h(s, X(s))$ such that

$$D_2 \log h(s, X(s))(Y) = \langle\nabla\log h(s, X(s)), Y\rangle_H. \tag{28}$$

Hence, we get the claimed value for $L(t)$.

Lastly, we find the quadratic variation of $L$. We can equivalently write $L(t)$ as

$$L(t) = \int_0^t \Phi(s)\mathrm{d}W(t) \qquad \Phi(s)(u) := \langle\nabla\log h(s, X(s)), B(X(s))u\rangle_H, \tag{29}$$

where $\Phi(t) \in HS(Q^{1/2}(U), \mathbb{R})$, i.e. the space of Hilbert-Schmidt operators from $Q^{1/2}(U)$ to $\mathbb{R}$. Then Röckner and Claudia [2007, Lemma 2.4.2] states that

$$\|\Phi(t)\|_{HS(Q^{1/2}(U),\mathbb{R})} = \|B^*(X(s))\nabla\log h(s, X(s))\|_{Q^{1/2}(U)}. \tag{30}$$

Finally, by Röckner and Claudia [2007, Lemma 2.4.3], it holds that

$$[L]_t = \int_0^t \|B^*(X(s))\nabla\log h(s, X(s))\|_{Q^{1/2}(U)}^2\mathrm{d}s. \tag{31}$$

$\square$

The proof of the theorem is now simply an application of Girsanov's theorem, normalising $Z$ by $\mathbb{E}[Z(T)]$ if necessary:

*Proof.* Let $\psi(s) \coloneqq B^*(X(s))\nabla \log h(s, X(s))$. Then $\psi$ is a $Q^{1/2}(U)$-valued $\mathcal{F}_t$ predictable process. By Lemma C.2 it holds that

$$Z(t) = \exp\left(\int_0^t \langle \psi(s), \mathrm{d}W(s)\rangle_{Q^{1/2}(U)} - \frac{1}{2}\int_0^t |\psi(s)|^2_{Q^{1/2}(U)}\mathrm{d}s\right) \tag{32}$$

Hence, applying Girsanov's theorem, we can define a new measure $\mathrm{d}\widehat{\mathbb{P}} \coloneqq Z(T)\mathrm{d}\mathbb{P}$, and know the Wiener process with respect to $\widehat{\mathbb{P}}$ has form

$$\widehat{W}(t) = W(t) - \int_0^t B(X(s))^*\nabla \log h(s, X(s))\mathrm{d}s. \tag{33}$$

Rewriting $X$ as a stochastic equation with respect to $\widehat{\mathbb{P}}$, we see that $X$ satisfies

$$\begin{aligned}X(t) =& X(0) + \int_0^t B(X(s))B(X(s))^*\nabla \log h(s, X(s))\mathrm{d}s \\ &+ \int_0^t [AX(s) + F(X(s))]\mathrm{d}s + \int_0^t B(X(s))\mathrm{d}\widehat{W}(s),\end{aligned} \tag{34}$$

giving the $h$-transformed process. $\qquad\square$

**Lemma C.3.** *(Lemma 5.2 in paper) Let $X$ be as in Equation* (1)*, satisfying Assumption 3.1. Given a function $h : [0, T] \times H \to \mathbb{R}$ satisfying Equation* (9)*, $Z(t) \coloneqq h(t, X(t, \xi_0))$ is a strictly positive martingale such that $Z(0) = 1$ and $Z(T) = C_T \cdot \psi(X(T, \xi_0))$.*

*Proof.* The functions of form $h(t, \xi)$ are Markov transition operators satisfying Equation (4). Define $Z(t) \coloneqq h(t, X(t))$. Then $Z(T) = \mathbb{E}[\psi(X(T, \xi_0)) \mid X(T, \xi_0)] = \psi(X(T, \xi_0))$. Hence, $\mathbb{E}[Z(T)] = 1$. Further, by the normalisation $C_T$, it holds $Z(0) = 1$. Strict positivity of $Z$ holds since $\psi$ is strictly positive. To see that $Z$ is a martingale, we use the tower property: let $Y$ be a random variable and $\mathcal{H}_1 \subset \mathcal{H}_2 \subset \mathcal{F}$. Then

$$\mathbb{E}[\mathbb{E}[Y \mid \mathcal{H}_2]|\mathcal{H}_1] = \mathbb{E}[Y \mid \mathcal{H}_1]. \tag{35}$$

Now, for $s < t$, $\mathcal{F}_s \subset \mathcal{F}_t$, we get

$$\mathbb{E}[Z(t) \mid \mathcal{F}_s] = \mathbb{E}[\mathbb{E}[C_T \cdot \psi(X(T, \xi)) \mid \mathcal{F}_t] \mid \mathcal{F}_s] = Z(s). \tag{36}$$

$\qquad\square$

**Lemma C.4.** *(Lemma 5.3 in paper) Let $\psi : H \to \mathbb{R}$ be a continuous function, twice Fréchet differentiable, with continuous derivatives. Then $h(t, \xi) \coloneqq \mathbb{E}[\psi(X(T - t, \xi))]$ is twice Fréchet differentiable in $\xi$ and once differentiable in $t$, with continuous derivatives.*

*Proof.* As before, we denote the $j^{\text{th}}$ Fréchet derivative with respect to the $i^{\text{th}}$ argument of a function $f$ by $D_i^j f$. If $f$ only has one argument then we will instead write $D^j f$. First note that by Assumption 3.2 $X(t, \xi)$ is twice Fréchet differentiable with respect to $\xi$ and has continuous derivatives. By our assumption that $\psi$ is twice Fréchet differentiable with continuous derivatives, we know that the composition $\psi(X(t, \xi))$ is also twice Fréchet differentiable with second derivative

$$\begin{aligned}D_2^2(\psi \circ X)(t, \xi)(h, g) =& \\ & D_2^2\psi(X(t, \xi))(D_2 X(t, \xi)h, D_2 X(t, \xi)g) \\ &+ D_2\psi(X(t, \xi))(D_2^2 X(t, \xi)(h, g)).\end{aligned} \tag{37}$$

This is continuous in $[0, T] \times H$. By Lebesgue's dominated convergence theorem and the definition of Fréchet differentiability, and noting that this holds for any $t \in [0, T]$, $h(t, \xi) \coloneqq \mathbb{E}[\psi(X(T - t, \xi))]$ is also twice Fréchet differentiable.

Next, we show that we can differentiate with respect to $t$. By Itô's lemma and properties of expectation, it holds that:

$$
\begin{aligned}
g(t,\xi) :=& \mathbb{E}[\psi(X(t,\xi))] \\
=& \psi(\xi) + \mathbb{E}\int_0^t (D\psi)(X(s,\xi))(AX(s,\xi) + F(X(s,\xi)))\mathrm{d}s \\
& + \frac{1}{2}\mathbb{E}\int_0^t \mathrm{Tr}_{B(X(s,\xi))Q^{1/2}}(D^2\psi)(X(s,\xi))\mathrm{d}s.
\end{aligned}
\tag{38}
$$

Note that by assumed continuity properties of $X$ and $\psi$, Equation (38) is continuous. Therefore, using Lebesgue's dominated convergence theorem and the fundamental theorem of calculus, we see

$$
\begin{aligned}
D_1 g(t,\xi) =& \lim_{r\to 0}\frac{1}{r}(g(t+r,\xi) - g(t,\xi)) \\
=& \lim_{r\to 0}\frac{1}{r}\mathbb{E}\int_t^{t+r}(D\psi)(X(s,\xi))(AX(s,\xi) + F(X(s,\xi)))\mathrm{d}s \\
& + \frac{1}{2}\mathbb{E}\int_t^{t+r}\mathrm{Tr}_{B(X(s,\xi))Q^{1/2}}(D^2\psi)(X(s,\xi))\mathrm{d}s \\
=& \mathbb{E}\left[(D\psi)(X(t,\xi))(AX(t,\xi) + F(X(t,\xi)))\right] \\
& + \frac{1}{2}\mathbb{E}\,\mathrm{Tr}_{B(X(t,\xi))Q^{1/2}}(D^2\psi)(X(t,\xi)),
\end{aligned}
$$

(39)

(40)

and so $g(t,\xi)$ is differentiable with respect to $t$. Noting that $h(t,\xi) = g(T-t,\xi)$, and $t \to T-t$ is differentiable, we get the result.

$\square$

**Lemma C.5.** *(Lemma 5.4 in paper) Let $\Gamma_N$ be as in Equation* (16) *and $h : [0,T] \times H^N \to \mathbb{R}$ be defined by $h(t,Y) := \mathbb{E}[\delta_{\Gamma_N}(X(T-t,Y))]$. Moreover, define $g : [0,T] \times \mathbb{R}^N \to \mathbb{R}$ by $g(t,y) := \mathbb{E}[\prod_{i=1}^N \delta_{\Gamma_i}(X_i^N(T-t,y))]$. Then $\langle \nabla \log h(t,Y), e_i \rangle = [\nabla \log g(t,(Y_i)_{i=1}^N)]_i$.*

*Proof.* First note that for any $Y \in H$,

$$
h(t,Y) = \mathbb{E}[\delta_{\Gamma_N}(X(T-t,Y))]
\tag{41}
$$

$$
= \mathbb{E}[\prod_{i=1}^N \delta_{\Gamma_i}(\langle X(T-t,Y), e_i\rangle)]
\tag{42}
$$

$$
= \mathbb{E}[\prod_{i=1}^N \delta_{\Gamma_i}(X_i^N(T-t,(Y)_{i=1}^N))] = g(t,(Y_i)_{i=1}^N).
\tag{43}
$$

Now note that for any $\varepsilon \in H^N$

$$
\frac{h(t,Y+\varepsilon) - h(t,Y)}{\|\varepsilon\|_{H^N}} = \frac{g(t,(Y_i)_{i=1}^N + (\varepsilon_i)_{i=1}^N) - g(t,(Y_i)_{i=1}^N)}{\|(\varepsilon_i)_{i=1}^N\|_{\mathbb{R}^N}}.
\tag{44}
$$

Hence, by the properties of the Fréchet derivative, it holds $\langle \nabla \log h(t,Y), e_i \rangle = [\nabla \log g(t,(Y_i)_{i=1}^N)]_i$.

$\square$

# D  Further background

## D.1  Infinite dimensional Itô and Girsanov

In order to condition, we rely heavily on the infinite-dimensional analogues of Itô's lemma and Girsanov's theorem. We state the exact version of both that we use.

Girsanov's theorem [Da Prato and Zabczyk, 2014, Section 10.2.1] allows us to define a change or reweighting of the measure and also tells us what stochastic processes look like with regard to this

new measure. Hence we can use this to get a change of measure which possesses some wanted behaviour and then write stochastic processes with respect to this measure. If we have a martingale $Z$ with respect to a probability measure $\mathbb{P}$, then we define a new probability measure $\mathrm{d}\widehat{\mathbb{P}} := Z(T)\mathrm{d}\mathbb{P}$, and we can write the $Q$-Wiener process of $\widehat{\mathbb{P}}$ in terms of the original Wiener process.

**Theorem D.1.** *Let $W$ be a $Q$-Wiener process in $U$ and let $\psi(\cdot)$ be a $Q^{1/2}(U)$-valued $\mathcal{F}_t$-predictable process. If*

$$Z(t) := \exp\left(\int_0^t \langle \psi(s), \mathrm{d}W(s)\rangle - \frac{1}{2}\int_0^t |\psi(s)|^2 \mathrm{d}s\right), \tag{45}$$

*where the inner product and norm are in the Hilbert space $Q^{\frac{1}{2}}(U)$, then $\mathbb{E}[Z(T)] = 1$. Then*

$$\widehat{W}(t) = W(t) - \int_0^t \psi(s)\mathrm{d}s \quad t \in [0, T] \tag{46}$$

*is a $Q$-Wiener process with respect to $\{\mathcal{F}_t\}_{t \geq 0}$ on $(\Omega, \mathcal{F}, \widehat{\mathbb{P}})$ for $\mathrm{d}\widehat{\mathbb{P}} = Z(T)\mathrm{d}\mathbb{P}$.*

Another theorem we shall be relying on is the infinite-dimensional analogue of Itô's lemma. This is the analogue of the chain rule for Hilbert space-valued processes. The version we use is adapted from a more general version for Banach spaces [Brzezniak et al., 2008, Theorem 2.4]. Da Prato et al. [2019] discuss Itô's lemma for Hilbert space-valued SDEs.

**Theorem D.2.** *Let $H$ and $E$ be separable Hilbert spaces. Assume that $f : [0, T] \times H \to E$ is of class $C^{1,2}$. Let $\Phi : [0, T] \times \Omega \to \mathrm{HS}(Q^{1/2}(U), H)$ be measurable and stochastically integrable with respect to $W$, a $Q$-Wiener process on $U$. Let $\psi : [0, T] \times \Omega \to H$ be measurable and adapted with paths in $L^1(0, T; H)$ almost surely. Let $\xi_0 : \Omega \to H$ be $\mathcal{F}_0$-measurable. Define $X : [0, T] \times \Omega \to H$ by*

$$X(t) = \xi_0 + \int_0^t \psi(s)\mathrm{d}s + \int_0^t \Phi(s)\mathrm{d}W(s). \tag{47}$$

*Then almost surely for all $t \in [0, T]$,*

$$\begin{aligned}
f(t, X(t)) = f(0, \xi_0) &+ \int_0^t D_1 f(s, X(s))\mathrm{d}s + \int_0^t D_2 f(s, X(s))\psi(s)\mathrm{d}s \\
&+ \frac{1}{2}\int_0^t \mathrm{Tr}_{\Phi(s)Q^{1/2}} D_2^2 f(s, X(s))\mathrm{d}s + \int_0^t D_2 f(s, X(s))\Phi(s)\mathrm{d}W(s).
\end{aligned} \tag{48}$$

*where $\mathrm{Tr}_{\Phi(s)Q^{1/2}} D_2^2 f(s, X(s))$ is defined as*

$$\sum_{j \geq 1} D_2^2 f(s, X(s))\left(\Phi(s)Q^{1/2}u_j, \Phi(s)Q^{1/2}u_j\right), \tag{49}$$

*for an orthonormal basis $\{u_j\}_{j \geq 1}$ of $U$.*

