# OpenReview forum: "Conditioning non-linear and infinite-dimensional diffusion processes"
_NeurIPS.cc/2024/Conference — NeurIPS 2024 spotlight_

### Official Review · Reviewer_veYd · 2024-07-10

**Soundness:** 3
**Presentation:** 2
**Contribution:** 3
**Rating:** 7
**Confidence:** 4

**Summary:**

The paper attempts to derive a means of conditioning a nonlinear diffusion process upon function-valued observations, via the $h$-transforms, adapting the method of _Jeremy Heng, Valentin De Bortoli, Arnaud Doucet, and James Thornton. Simulating diffusion bridges with score matching._ to a function-valued setting.

**Strengths:**

Discrete approximations of notionally continuous objects is a ubiquitous problem in machine learning. By representing conditional nonlinear SDE solutions themselves in function space, this expands the range and type of discretization that can be employed to solve problems which are naturally regarded as functions; in this paper, it enables the use of reasonably general  (separable) Hilbert-space basis functions as the means of discretization, rather than, e.g. a raster grid.

This problem is interesting and well-posed.

**Weaknesses:**

There are many small oddities in the style which make this paper a difficult read.

See below for those.

The paper presents essentially one result, which is the up-lifting of learned bridge diffusion on a finite dimensional vector space, to ones on a function space with a finitely-truncated basis. This result seems somewhat, if not massively, important.

The first two pages, before the problem statement, are confusing. If we read the paper in linear order we cannot understand many of the assertions made there without reference to equations which have not been introduced yet, and are not even cross referenced.  e.g. l46/sect 2.1

>Given an SDE, the conditioned SDE contains an intractable score function. This is similar, but46
slightly different, to the score function that arises in generative diffusion models Vincent [2011],47
Song and Ermon [2019], Song et al. [2021]. There, the starting distribution is complicated, but the48
stochastic process is linear. In our case, we are interested in the process itself, particularly non-linear49
processes. In this way, our work generalises the finite-dimensional work on conditioning non-linear50
SDEs and infinite dimensional score matching, where they consider time reversals of linear SDEs

What is doing on? Which score function is intractable? There is a lot of this kind of thing where technical statements are made without reference to the equations that ground them. This becomes (more) clear after reading the whole paper, but in the order that the paper is written, this entire section is hard to parse.

**Questions:**

1. What is happening in figure 1? It is not easy to parse the start and end shapes. I can just about work it out from section 6.2.2, but can we add some visual cues in the figure, for example, fading out the starting shape as time does on, and fading in the terminal one?

   > we first use two butterflies with somewhat320 different shapes [GBIF.Org User, 2024]. One trajectory between the two butterflies is   plotted in Figure 2. In this, we can see the high correlation between neighbouring points, with a Brownian temporal model. In Figure 1, we plot 120 butterfly trajectories, at specific time points. For t = 0.2 we see that the butterfly outlines are mostly close to the start butterfly in pink, and at time t = 0.8, they are closer to the green target butterfly

2. 4.2/l159

   > Moreover, under this measure $\mathbb{Q}, x(t)$ satisfies a new SDE
   >$$
   > \mathrm{d} x^c(t)=f\left(t, x^c(t)\right) \mathrm{d} t+\sigma \sigma^T\left(t, x^c\right) \nabla_x \log h\left(t, x^c(t)\right) \mathrm{d} t+\sigma\left(t, x^c(t)\right) \mathrm{d} W(t) .
   >$$

   Can you clarify the relationship between $x$ and $x^c$?

3. l165 confusing phrasing

   > When $h(t, x)=p(t, x ; T, y)$ there is no general closed form solution. Different methods to learn the bridge exist Delyon and Hu [2006], Schauer et al. [2017]. More recently, score-based learning methods were proposed to learn the term $\nabla_x \log p(t, x ; T, y)$ Heng et al. [2021].

   Do you mean something like this?

   > For the required  $h(t, x)=p(t, x ; T, y)$ there is no general closed form for $h$. Different methods to learn the bridge exist Delyon and Hu [2006], Schauer et al. [2017]. More recently, score-based learning methods were proposed to learn the term $\nabla_x \log p(t, x ; T, y)$ Heng et al. [2021], and it is the infinite-dimensional generalisation of the latter method  that we pursue here

4. eq15:

   >$\left\langle X(t, \xi), e_i\right\rangle=\left\langle\xi, e_i\right\rangle+\int_0^t\left\langle A X(s)+f(X(s)), e_i\right\rangle \mathrm{d} t+\int_0^t\left\langle e_i, B(X(s)) \mathrm{d} W(s)\right\rangle$.

   Is something wrong with the variables of integration here? $\int_0^t\left\langle A X(s)+f(X(s)), e_i\right\rangle \mathrm{d} t$ is an integral in $t$ and yet the integrand doesn't depend upon t, and it does depend upon $s$ which is a free variable

Minor typos:

* l195

  > However, transition operators of form Sec. 4.1 exist and satisfy the Markov property Equation (4)

  should that be

  > However, transition operators of form Equation (3) exist and satisfy the Markov property Equation (4)

**Limitations:**

The paper seems to depend upon explicit orthogonal bases (sec 5.3) which is IMO a  restriction in practice, since the diffusion methods of industrial interest frequently have no such explicit basis.

---

> ### Author Rebuttal · Authors · 2024-08-06
>
> Thanks for your positive review and the concrete suggestions. We're glad you appreciate the importance of our result!
>
> It seems to us that the weaknesses you listed were almost entirely presentational. We have fixed the issues you point out in your review as follows.
>
> **Clarity of introduction and related work:**
>
> Thank you for your feedback on the first two pages! We have now edited these sections, some examples of which we give in the following.
>
> > Given an SDE...
>
> Thanks for pointing out this paragraph! This section states that for given $T, y$ the score function $\nabla \log p(t, x; T, y)$ is intractable, since for nonlinear SDEs there is no known closed solution for $p(t, x; T, y)$.
> We have now removed this paragraph and instead incorporate it into the related work on infinite dimensional diffusion models. The corresponding subsection in the related work now reads (previously lines 69-74):
>
> > Recent work on generative modelling has investigated score matching for infinite-dimensional diffusion processes [Pidstrigach et al., 2023, Franzese et al., 2023, Bond-Taylor and Willcocks, 2023, Hagemann et al., 2023, Lim et al., 2023]. This problem is similar to our task of conditioning an SDE, but not the same: The main difference is that our SDEs are fixed, known a-priori, and potentially nonlinear, whereas in generative modelling the SDE can be chosen freely. Hence, generative modelling often uses linear SDEs because the transition densities are known in closed form. In this sense, our problem relates to generative modelling, but has a different setup.
>
>
> The references to generative diffusion models have been moved to the paragraph about score matching for finite-dimensional nonlinear bridges (previously lines 61-68):
>
> > Recently, Heng et al. [2021] adapted the score-matching methods of Vincent (2011), Song and Ermon (2019), and Song et al. (2021) to learn the score term for non-linear bridge processes. To do so, they introduce a new loss function to learn the time reversal of the process. They then learn the time reversal of the time reversal, which gives the forward bridge. Our work uses their method to learn the score term after discretising the SDE via truncated sums of basis elements. Phillips et al. [2022] also consider using truncated sums of basis elements for discretising SDEs, however, only for infinite-dimensional Ornstein-Uhlenbeck processes, which are linear.
>
>
> We hope that these changes resolve your presentational issues, and that you agree with us that the presentation is improved.
>
>
> **To answer your questions:**
>
> 1. Thanks for the suggestion. We've tried to make this figure clearer now (see PDF in the general reply). The caption in the PDF will also be included in the paper, which we hope gives additional clarity.
>
> 2. Thanks for the question. The process $x^c$ can be thought of as the conditioned version of $x$. For example if $h(t, x) = \frac{p(t, x; T, y)}{p(0, x_0; T, y)},$ then for a set $B\subset\mathbb{R}^d$, $\mathbb{P}(x^c(t) \in B) = \mathbb{P}(x \in B \mid x(T)=y)$ holds.
> We've refined the corresponding explanation in Section 4.2.
>
> 3. Yes, this is precisely what we mean. We've edited it, so it's hopefully more clear now. The new version reads:
>
>     > For $h(t, x) := \frac{p(t, x; T, y)}{p(0, x_0; T, y)},$ as in conditioning on an end point, there is, in general, no closed form solution for $h$. Different methods to learn the bridge exist (Delyon and Hu [2006], Schauer et al. [2017]). More recently, score-based learning methods were proposed to learn the term $\nabla_x \log p(t, x; T, y)$ (Heng et al. [2021]), which we will adapt to the infinite-dimensional setting.
>
> 4. That was a typo, thanks for catching it! It is supposed to read
>     $$\langle X(t, \xi), e_i \rangle = \langle \xi, e_i\rangle + \int_0^t \langle AX(s) + f(X(s)), e_i\rangle \mathrm{d}s + \int_0^t\langle e_i, B(X(s))\mathrm{d}W(s)\rangle.$$
>     We have corrected this typo in the paper.
>
> We would also like to clarify the limitation you mentioned. There are some non-standard Hilbert spaces without explicit orthogonal bases (e.g. Sobolev spaces on non-standard manifolds). However, when working in $L^2$ or Sobolev spaces on Euclidean spaces and spheres, one does have access to explicit bases, so this might be less of a limitation than indicated in the review.
>
> We hope that you agree that the presentational aspects are now improved!
> In any case, thank you for the positive evaluation and the concrete suggestions for how to improve the presentation -- we've updated the paper accordingly, as outlined above.

---

> > ### Comment · Reviewer_veYd · 2024-08-11
> >
> > I thank the authors for their explanations. Indeed, my concerns are mostly presentational; I think this is a good paper. The authors have addressed my questions. I have revised my score up accordingly.

---

### Official Review · Reviewer_bXXx · 2024-07-10

**Soundness:** 3
**Presentation:** 4
**Contribution:** 3
**Rating:** 7
**Confidence:** 3

**Summary:**

This paper explores the conditioning of non-linear processes in infinite dimensions. To achieve this, the authors introduce an **infinite version of Doob’s $h$-transform** (contribution 1) that relies on the infinite-dimensional counterparts of Itô’s lemma and Girsanov’s theorem. They then discretize the conditioned process and use score-matching techniques to **learn the score** arising from the $h$-transform by training on the coefficients of the Fourier basis, which allows sampling from the conditioned process (contribution 2).

These mathematical tools are used to condition a process to hit a specific set at the end time, also known as bridges. The authors  **detail two models** based on different scenarios: one for direct conditioning on data (**exact matching**) when the transition operator of the SDE solution is smooth, and the second for assuming some observation error (**inexact matching**) (contribution 3).

They **illustrate their procedure by modeling changes in the morphometry** (i.e., shapes) of organisms in evolution, specifically the changes in the shapes of butterflies over time (contribution 4).

**Strengths:**

The theoretical mathematical contribution, namely the conditioning of non-linear processes in infinite dimensions, is noteworthy and broadens the scope of previous work that focused on approximating non-linear bridge processes in finite dimensions (Delyon and Hu, 2006, van der Maulen and Schauer, 2022). The infinite version of Doob’s $h$-transform, although not surprising in its form (similar to the finite-dimensional case), is of independent interest. The general procedure, based on this transform, allows conditioning without discretizing the model beforehand.

Two models are developed: exact matching and inexact matching. Inexact matching involves conditioning the process so that at the final time it does not exactly satisfy a final condition but approaches it. This approach is particularly interesting as it incorporates potential observation errors (by introducing noise) and relaxes the restrictive Assumption 3.3, which is unavoidable in the case of exact matching.

The application to modeling changes in morphometry, using bridge processes between shapes, is highly relevant for illustrating the usefulness of the developed procedure. Unlike previous work on the subject (Arnaudon et al., 2019, 2022), the conditioning precedes discretization, ensuring the proper definition of the bridge even as the number of points tends to infinity.

The overall presentation of the paper is excellent: the introduction effectively situates the study within the existing literature on related topics (approximation of non-linear bridge processes, learning score functions in generative diffusion models, diffusion bridges in shape spaces), the contributions are clearly outlined, and the tools developed (the infinite version of Doob’s $h$-transform) are introduced in a pedagogical and concise manner without sacrificing rigor.

Apart from a few minor confusions in the notation, the proofs seem correct and well-written.

**Weaknesses:**

### Assumption 3.3

As mentioned in the article itself, Assumption 3.3—indispensable in the case of exact matching—concerning the regularity of the transition function, is strong. An example of a subset $ \Gamma$ (finite-dimensional cylinder) matching this condition is provided in Section 5.3. It seems to me that the inherent difficulty of Assumption 3.3 for exact matching in infinite dimension is circumvented by choosing an example where the problem is ultimatly 'reduced to finite dimension'. Maybe having an example with conditions on the solution process itself $(X_t)\_{t\in\mathbb{R}_+}$ or the coefficients of the SDE it satisfies, which illustrate assumption 3.3, would be more interesting.



### Theoretical achievement

As it stands, the article develops an interesting method (though perhaps not completely groundbreaking) based on an extension of the Doob $h$-transform and reversing the usual discretization-conditioning steps, allowing for a well-defined bridge despite the difficulty associated with infinite dimensions. Perhaps a theoretical study of the error between  the conditioning and the true solution would strengthen relevance of the approach.

### Notation

It's not really a weakness, but the notations should be harmonized (for example $ D_2 $ or $D_x$, $x_0$ or $\xi_0$) to make reading and reviewing the proofs easier. Perhaps a summary table of notations could be included?

**Questions:**

The questions follow the potential identified weaknesses:

**Assumption 3.3** Can you illustrate it by providing conditions on the solution process $(X_t)\_{t\in\mathbb{R}_+}$ itself rather than on the set $\Gamma$? Perhaps using results on the regularity of the solution process density via the Malliavin derivative (S. Kusuoka and D. Stroock, "Applications of Malliavin calculus, part II", Kohatsu-Higa and Tanaka, Annales IHP 2012, D. Nualart, M. Zakai, Séminaire de probabilités 1989), or the parametrix method (Bally and Kohatsu-Higa, AAP 2015)?

**Theoretical analysis** Would it be possible to quantitatively measure the quality of the procedure, i.e.  to provide an upper bound on the error between the conditioning and the true solution in the case of exact matching? In the case of inexact matching, can we measure the impact of the noise on this error?


**Proof of Lemma C.4** It seems to me that the proof of Lemma C.4 corresponds to the calculation of the infinitesimal generator associated with the process
$(X_t)\_{t\in\mathbb{R}_+}$ and not to what is stated. In the statement of the Lemma $h$ is defined as $h(t,\xi)=\mathbb E[\psi(X(T-t,\xi))]$ whereas in the proof, $h(t,x):=\mathbb E[\psi(X(t,\xi))]$. Adapt the proof maybe by defining $g(t,x)=h(T-t,x)$.


### Minor comments

There are some typographical errors and notational awkwardness. These notation issues recur repeatedly:

- l. 98, 137, 185 etc.: Write $W$ or $\{W_t\}$ instead of $W_t$ when dealing with a process. Same remark for $e^{tA}$ (l. 101).
\item l. 140, 147, 198 etc.: The initial condition of the SDE is denoted first by $x_0$ (Equation (1)) and later by $\xi_0$, $x$. Please harmonize the notation.
- l. 590, Equations (35) and (36): Partial derivatives can be denoted as $D_x$ or $D_2$. Please harmonize the notation.

Here is a non-exhaustive list:
- l. 80: $f(T,s_0)=s_1,f$. $f$ and ? I don't understand the sentence/
- l. 155: Add $x(0)=x_0\in\mathbb{R}^d$.
- l. 156: $X$ should be lowercase.
- l. 158: What is $p$ ?
- l. 158: I think we should have $\mathbb{E}[Z(T)]=1$.
- l. 159: I think is it $d\mathbb{Q}/d\mathbb{P}|\mathcal F_t$.
- l. 234: Equation (13), what is $v$?
- l. 553: Define $[L]$.
- l. 558 : How do you defined $D_2$ ? I think it's $\partial_x$ or $h_x$ ?
- l. 562: Using that $Z(t)=h(t,X(t))$
- l. 569: What is $H_Q$ ?
- l. 583: $Z(s)$ instead of $Z_s$
- l. 583 : $C_T=1/P_T\psi(\xi)$ by definition
- l. 590 equation (35), equation (36) harmonisation of notation  with the reference lemma of Itô formula
- l. 599 : Lemma C.5, How do you define  $c^i$ and l. 601 $c_i$?
- l. 618 : $d\widehat P=Z(T)d\mathbb{P}$ ($d$ is missing)
- Define properly the Hilbert space $Q^{1/2}(H)$.

**Limitations:**

The limitations are briefly discussed in the conclusion, particularly the fact that the procedure would not be applicable to weak solutions of SDEs. Some directions for future research (focus on network architecture to increase the dimension that can be considered, infinite-dimensional bridges to inference problems) are also provided. There is no potential negative societal impact in this work.

---

> ### Author Rebuttal · Authors · 2024-08-06
>
> Thank you for the review and the insightful questions. We're glad you liked the paper and find the theoretical contribution noteworthy!
>
> In the following, we address your questions one by one:
>
> 1. Assumption 3.3: Yes, you're right and we agree this would be nice to include!
>     We are aware of a result for SDEs of form
>     $$dX = [AX + F(X)]\mathrm{d}t + \sqrt{Q}\mathrm{d}W(t),$$
>     with $F$ being once differentiable, and $A, Q$ satisfying some extra assumptions (see Theorem 9.39/9.43 of [1] or Section 6.5, 7.3 of [2]; references below). We will include this in the discussion of Assumption 3.3.
>     In general, it would indeed be nice to prove something for other cases, especially in the case of stochastic flows as in [3], which could perhaps be done using the Malliavin derivative as you point out. We plan on looking into this for future work.
>
> 2. Theoretical analysis: We agree this kind of result would be interesting! However, deriving such an error estimate would require too much additional analysis for now, which is why we leave it to future work.
>
> 3. Proof of lemma C.4: Yes, you're right! We've corrected this now. Concretely: We define $g(t,x)=h(T-t,x)$ (where we before showed that $g$ is differentiable in time). Then, note that the function $t \to T-t$ is differentiable, and therefore $h(t, x)$ is, too. The spatial Fréchet differentiability still holds, since we showed it holds for all $t \in [0, T]$.
>
> Thanks a lot for the feedback on the notation and for listing the small errors! We've now incorporated these into the paper and made sure all the notation is consistent.
> We hope that you agree that this improves the presentation.
>
> Thank you again for the positive review, and we look forward to further discussion!
>
> [1] Giuseppe Da Prato and Jerzy Zabczyk. Stochastic Equations in Infinite Dimensions. Encyclopedia of Mathematics and its Applications. Cambridge University Press, Cambridge, 2nd edition, 2014.
>
> [2] Cerrai, S., Second Order PDE’s in Finite and Infinite Dimension: a Probabilistic Approach, Lecture Notes in Mathematics. Springer, 2001.
>
> [3] Hiroshi Kunita. Stochastic Flows and Stochastic Differential Equations. Cambridge Studies in Advanced Mathematics. Cambridge University Press, Cambridge, 1st edition, 1997.

---

> > ### Comment · Reviewer_bXXx · 2024-08-09
> > **Acknowledgement of the rebuttal**
> >
> > I thank the authors for their rebuttal. I remain confident of the quality of their paper, suggest the acceptance and keep my score.

---

### Official Review · Reviewer_haqz · 2024-07-13

**Soundness:** 3
**Presentation:** 3
**Contribution:** 3
**Rating:** 7
**Confidence:** 1

**Summary:**

This paper addresses the challenge of conditioning infinite-dimensional stochastic processes, particularly non-linear ones, without prior discretisation. Traditional methods condition finite-dimensional data but struggle with infinite-dimensional, function-valued data. The authors employ an infinite-dimensional version of Girsanov’s theorem and Doob’s h-transform to condition such processes. This method is applied to time series analysis of shapes in evolutionary biology, specifically modelling changes in the morphometry of organisms. The paper also utilizes score matching techniques to learn the coefficients of the score function in the Fourier basis.

**Strengths:**

- The paper introduces a novel method for conditioning infinite-dimensional non-linear processes without prior discretization, generalizing recent work on linear processes in infinite dimensions.
- The authors derive Doob’s h-transform for infinite dimensional non-linear processes, allowing conditioning without first discretizing the model. Then, score matching is used to learn the score arising from the h-transform by training on the coefficients of the Fourier basis.
- The paper demonstrates a practical application to evolutionary biology to model changes in the shapes of organisms.

**Weaknesses:**

- The empirical experiments focused specifically on modeling the change in the shape of butterflies. Thus, it's unclear how the method performs in more general benchmarks for diffusion processes.
- Computational complexity could be large for the the proposed method especially for the non-linear setting.
- The evaluation lacks necessary comparison with related approaches.

**Questions:**

What is the computational complexity of the method?

---

> ### Author Rebuttal · Authors · 2024-08-06
>
> Thanks for the positive review!
> In the following, we would like to briefly clarify the lack of related approaches (this work is, to the best of our knowledge, the first to operate in an infinite-dimensional, nonlinear setting)
> and describe the computational complexity:
>
> 1. Comparison with related approaches / benchmarks: We are unaware of benchmarks respectively other approaches for nonlinear **and** infinite-dimensional processes -- we are only familiar with nonlinear (but finite-dimensional) (e.g. [1]) as well as infinite-dimensional (but linear) comparisons (e.g. [2]), neither of which directly compare to our work.
>
> 2. Complexity: Thanks for asking! The complexity of a single evaluation of the loss of a minibatch of $B$ trajectories with $N$ time steps and in dimension $d$ (which in this case corresponds to the number of basis elements) is $O(B \cdot N \cdot d^3)$. The cubic factor comes from covariance-matrix arithmetic, which is common for multi-output stochastic processes (both finite- and infinite-dimensional) (see e.g. [3]). We are planning on bringing that factor down in a follow-up project.
>
> We thank you again for your positive evaluation! We hope that we were able to clarify a few points and we look forward to the discussion!
>
> [1] Frank van der Meulen and Moritz Schauer. Automatic backward filtering forward guiding for markov processes and graphical models. arXiv preprint arXiv:2010.03509, 2022.
>
> [2] Jakiw Pidstrigach, Youssef Marzouk, Sebastian Reich, and Sven Wang. Infinite-dimensional diffusion models for function spaces. arXiv preprint arXiv:2302.10130, 2023.
>
> [3] James Hensman, Nicolò Fusi, and Neil D. Lawrence. Gaussian processes for Big data. Proceedings of the Twenty-Ninth Conference on Uncertainty in Artificial Intelligence. 2013.

---

> > ### Comment · Reviewer_haqz · 2024-08-11
> > **After rebuttal**
> >
> > The authors addressed my comments, and I've raised my score accordingly.

---

### Author Rebuttal · Authors · 2024-08-06

We thank all reviewers for their reviews and positive evaluations of our work. We are glad you all liked our contribution!

Many weaknesses seem to relate to presentational concerns, which we believe are easy to correct.
Below, we reply to all reviews in separate threads.
Attached is a PDF that contains an update to Figure 1, relating to the review by Reviewer veYd.

We look forward to the discussion!

Thank you again for the positive reviews, and best wishes,

The authors

---

### Decision · Program_Chairs · 2024-09-25

**Decision:**

Accept (spotlight)

**Comment:**

The reviewers agree that this is an interesting paper that deserves publication. There were a number of concerns about presentation that the authors should carefully address in the final version. A weakness of this paper is the empirical section, that focuses on a single task (modeling changes in shapes of butterflies)